# An Evaluation of the Impact of Air Pollution on the Lung Functions of High School Students Living in a Ceramic Industrial Park Zone

**DOI:** 10.3390/ijerph20216964

**Published:** 2023-10-24

**Authors:** Rafael Futoshi Mizutani, Ubiratan de Paula Santos, Renata Ferlin Arbex, Marcos Abdo Arbex, Mario Terra-Filho

**Affiliations:** 1Pulmonary Division, Heart Institute (InCor), Hospital das Clinicas da Faculdade de Medicina da Universidade de Sao Paulo, Sao Paulo 05403-000, Brazil; 2Independent Researcher, Araraquara 14801-534, Brazil; rfarbex@hotmail.com; 3Faculdade de Medicina, Universidade de Araraquara, Sao Paulo 14801-340, Brazil

**Keywords:** air pollution, environmental exposure, particulate matter, silicon dioxide, respiratory function tests, adolescent health

## Abstract

Santa Gertrudes (SG) and Rio Claro (RC), Sao Paulo, Brazil, are located in a ceramic industrial park zone, and their particulate matter with an aerodynamic diameter of less than 10 µm (PM_10_) concentration levels has been among the highest in recently monitored cities in Brazil. Local PM_10_ was mostly composed of silica. A cross-sectional study was designed to evaluate the lung functions of public high school students in SG, RC, and São Pedro (SP) (control location), Brazil, in 2018. The prevalence of asthma, mean PM_10,_ FVC (forced vital capacity), and FEV_1_ (forced expiratory volume in the first second) were compared between the locations, and regression analyses were performed. A total of 450 students were included (SG: 158, RC: 153, and SP: 139). The mean FVC% (SG: 95.0% ± 11.8%, RC: 98.8% ± 12.9%, SP: 102.4% ± 13.8%, *p* < 0.05), the mean FEV_1_% (SG: 95.7% ± 10.4%, RC: 99.7% ± 12.0%, SP: 103.2% ± 12.0%, *p* < 0.05) and the mean PM_10_ (SG: 77.75 ± 38.08 µg/m^3^, RC: 42.59 ± 23.46 µg/m^3^, SP: 29.52 ± 9.87 µg/m^3^, *p* < 0.01) differed between locations. In regression models, each increase in PM_10_ by 10 µg/m^3^ was associated with a decrease in FVC% by 1.10% (95% CI 0.55%–1.65%) and a decrease in FEV_1_% by 1.27% (95% CI 0.75%–1.79%). Exposure to high levels of silica-rich environmental PM_10_ was found to be associated with lower FVC and FEV_1_.

## 1. Introduction

Outdoor air pollution is a major health risk factor in childhood, associated with several lung diseases, such as asthma [1] and respiratory infections [2]. Long-term exposure to traffic-related air pollution during childhood is associated with impaired lung development. Children and adolescents living in areas with high levels of air pollutants, such as particulate matter with an aerodynamic diameter of less than 10 µm (PM_10_), particulate matter with an aerodynamic diameter of less than 2.5 µm (PM_2.5_), nitrogen oxides, black carbon, and acid vapor, have a higher risk of low forced expiratory volume in the first second (FEV_1_) and forced vital capacity (FVC) [3,4].

Santa Gertrudes (Sao Paulo, Brazil) (SG) is located in a ceramic industrial park zone that also extends to the neighboring town of Rio Claro (Sao Paulo, Brazil) (RC). PM_10_ concentration levels in SG and RC have been among the highest in monitored cities in Brazil during the past few years [5,6]. The mineralogical evaluation of atmospheric PM_10_ samples collected in the area revealed that PM_10_ is mostly composed of silica particles (up to 89.0% of the mass, measured using energy-dispersive spectroscopy in an atmospheric sample of PM_10_) [7], and its main sources are soil dust suspension due to heavy vehicle traffic on unpaved roads and open-air ceramic industrial activities [7]. Few studies evaluate the health effects on people living near industrial parks with high emissions of silica-rich air pollution [8,9,10], but there are no data on its effects on lung function in the exposed population.

Our hypothesis was that adolescents living in areas of high silica-rich PM_10_ have lower FVC and FEV_1_ than those living in areas of low PM_10_.

## 2. Materials and Methods

This cross-sectional study evaluated the lung function of adolescents living for the past 10 years in the towns of SG, RC, and São Pedro (SP) in Sao Paulo State, Brazil, in 2018 (Figure 1).

The study sample size calculation was based on the study of Gauderman et al. [3] and PM_10_ data from the Environmental Agency of Sao Paulo State (CETESB) [2]. The estimated prevalence of students with FEV_1_ below the lower limit of normality (LLN) in the most and the least polluted locations was, respectively, 9.5% and 1.9%, which was estimated as 20% higher than the prevalence data of the study by Gauderman et al. [3] due to higher PM_10_ levels in SG. A sample size of 145 participants was calculated for each location, using a type I error of 0.05 and a power of 0.8.

### 2.1. Procedures

Regional offices of the Sao Paulo State Secretary of Education were contacted, which provided access to selected public high schools at SG, RC, and SP. Students between 15 and 19 years living for the past 10 consecutive years in the same location were invited to participate in the study. Parental and/or legal guardian consent was obtained for underage students (less than 18 years of age), and all students were required to assent to study participation. This study was approved by the University of Sao Paulo Research Ethics Committee (protocol number 2.728.826).

The students were evaluated in their respective schools by research staff. They underwent a clinical evaluation, which included a structured questionnaire, a smoking status questionnaire, weight and height measurements, the International Study on Asthma and Allergies in Childhood (ISAAC) asthma questionnaire, and spirometry. Exclusion criteria included being under 15 or above 19 years of age, unable to perform spirometry, and not living in the same location for the past 10 years. The latter intended to select students who were most exposed to the same levels of air pollution in their lifetime, thus minimizing the chance of inclusion of subjects who were exposed to low levels of air pollution for most of their lifetime living at the time of the study at a high-pollution location and vice versa.

The ISAAC questionnaire is a well-established tool for screening asthma, rhinitis, and atopic dermatitis symptoms in population-based studies. The tool includes a validated Portuguese translation for the asthma questionnaire [11].

Spirometry tests were performed by trained technicians using portable spirometers (Koko^®^ Sx, Nspire Health Inc., Longmont, CO, USA) that were calibrated once daily with a 3 L calibration syringe. The inability to perform spirometry was defined as less than two acceptable, usable, and repeatable maneuvers in an upper limit of eight attempts [12]. The Global Lung Initiative Network’s (GLI) spirometry reference values were used [13]. The tests were interpreted by a single pulmonologist according to the 2021 ATS/ERS recommendations [14].

### 2.2. Disease Definition

Childhood obesity and malnutrition were defined using the 2007 World Health Organization (WHO) reference data on body mass index-for-age (BMI-for-age) between 5 and 19 years, in which obesity was defined as a BMI-for-age Z-score ≥ +2, and malnutrition was defined as a BMI-for-age Z-score ≤ −2 [15]. Asthma was defined as an ISAAC asthma questionnaire score (AS) ≥ 6 [7] plus evidence of airflow obstruction on spirometry [11,16].

### 2.3. Air Pollution Data

PM_10_ data have been regularly monitored by CETESB fixed samplers in SG since 2007 and RC since 2011; however, they have not been monitored in SP, as the location is a small tourist resort town with no relevant industrial activities. PM_10_ measurements were obtained for SG and RC in 2018 using the CETESB database [5]. PM_10_ measurements were retrieved in SP with a portable particulate matter monitor (Dusttrak^TM^ II Aerosol Monitor 8530, TSI Inc., Saint Paul, MN, USA) during the evaluation days. PM_10_ was measured using CETESB with manual weighting collected over 24 h via filtration using gravimetric sampling every six days [17]. Dusttrak^TM^ II measured PM_10_ using a 90º light-scattering sensor with gravimetric sampling every 60 s [18].

### 2.4. Statistical Analysis

Descriptive characteristics of the evaluated students were compared by location.

Spirometry data were compared by location with prevalence rates of obstructive or non-specific patterns, prevalence rates of FVC% (forced vital capacity percent of predicted), FEV_1_% (forced expiratory volume in first-second percent of predicted) below the lower limit of normality (LLN), below 80% of predicted, below 90% of predicted, and mean FVC% and FEV_1_%.

The mean PM_10_ concentration in 2018 was calculated for SG and RC, and the mean PM_10_ concentration during the evaluation days in SP.

Normally distributed continuous data were compared using analysis of variance (ANOVA) with Tukey’s test for pairwise post hoc analysis. Categorical data were compared using the chi-squared or Fisher’s test, with a pairwise post hoc analysis with a Bonferroni correction.

The Pearson correlation (r) was performed between the prevalence of FVC below the LLN and PM_10_ and between the prevalence of FEV_1_ below the LLN and PM_10_.

Univariate linear regression analyses were performed using FVC% and the following variables: BMI, obesity, malnutrition, smoking status (never smoker vs. former/current smoker), AS ≥ 6, asthma diagnosis, and PM_10_ level. The same variables were analyzed using FEV_1_%. Variables with a linear regression with a *p*-value < 0.1 were selected to build multiple linear regression analysis models. Only one among the variables that evaluated a similar parameter was chosen to build the multiple linear regression model (e.g., only one variable among BMI, obesity, and malnutrition would be chosen).

All statistical analyses were performed using R v. 4.03 (R Development Core Team, Auckland, New Zealand). Statistical significance was defined based on a *p*-value less than 0.05 in two-tailed tests.

## 3. Results

A total of 507 students were evaluated in four schools (two in SG). SG evaluations occurred between April 17 and 20 April 2018 and 19 and 20 July 2018. SP evaluations occurred between 6 and 8 August 2018. RC evaluations occurred between 8 and 10 October 2018. A total of 57 students were excluded for the following reasons: 18 were under 15 or above 19 years of age, 17 were living for less than 10 years in the same location, and 20 were unable to perform spirometry. In total, 450 students were included in the analysis.

The students living in SP were older than those living in SG. The students living in RC had a higher mean BMI and prevalence of obesity than those living in SG. The groups did not differ in other characteristics, including the prevalence of asthma symptoms and diagnosis (Table 1).

The students living in SG had a lower mean FVC% and FEV_1_% than those living in RC and SP (*p* < 0.05), and students living in RC had a lower mean FVC% and FEV_1_% than those living in SP (*p* < 0.05). However, there were no differences in the prevalence of FVC or FEV_1_ below the LLN. Using a threshold of 80% produced similar results, with a statistically significant chi-squared test for FVC, but without differences in the pairwise post hoc analysis with a Bonferroni correction. Using a threshold of 90%, the prevalence of FVC < 90% was higher in SG than in SP cohorts (*p* < 0.05), and the prevalence of FEV_1_ < 90% was higher in SG than in RC and SP cohorts (*p* < 0.05) (Table 2).

The mean PM_10_ concentrations were higher in SG than in RC and SP cohorts and were higher in RC than in SP (*p* < 0.01) (Table 3). PM_10_ highly correlated with the prevalence of FEV_1_ below the LLN (r = 0.99 and *p* = 0.02) and of FVC below the LLN (r = 0.98 and *p* = 0.09) (Figure 2).

The univariate regression model analysis showed an association between FVC% and BMI, malnutrition, obesity, and PM_10_. FEV_1_% was associated with BMI, malnutrition, PM_10_, AS ≥ 6, and asthma diagnosis (Table 4). Multiple linear regression models were built using FVC%, BMI, and PM_10_ and using FEV1%, BMI, PM_10_, and an asthma diagnosis. After correction, each increase in PM_10_ by 10 µg/m^3^ was associated with a decrease in FVC% by 1.10 percentage points (CI 95% 0.55–1.65) and a decrease in FEV_1_% by 1.27 percentage points (CI 95% 0.75–1.79) (Table 5).

## 4. Discussion

Our original study contributes to the literature on the effects of air pollution on Latin American populations. Adolescents living in the most polluted locations had lower pulmonary function parameters. Higher PM_10_ was associated with lower mean FVC% and FEV_1_%, and each increase in PM_10_ by 10 µg/m3 was associated with a decrease in FVC% by 1.10% (95% CI 0.55–1.65) and in FEV_1_% by 1.27% (95% CI 0.75–1.79). The prevalence of FVC and FEV_1_ below the LLN did not differ between locations but correlated with PM_10_ concentration.

Most studies that evaluated the impact of air pollution on lung function in children and adolescents have found that populations exposed to air pollutants have lower lung function parameters, with a tendency for FEV_1_ to be more affected than FVC (15). In a review of the literature, Schultz et al. [19] found that most published studies on this subject were cross-sectional with only one spirometry measurement (32/44), few were longitudinal cohorts (12/44), and almost all (41/44) were conducted in North America or Europe, with only one conducted in Latin America (Mexico) [20].

We did not find differences in the prevalence of asthma diagnoses between locations. Although evidence of the causality of air pollution on incident asthma in childhood is increasing [1], paradoxically, studies that evaluated the association between air pollution and asthma prevalence in children/adolescents have found no or little association due to the heterogeneity of asthma definitions and types of air pollution analyses [21,22,23]. In our study, the prevalence of asthma was lower (2.4%) than the literature estimates (15–51%) [11,21], probably due to our stricter definition of asthma that required both the ISAAC questionnaire above a threshold and the presence of airflow obstruction in spirometry. If only the ISAAC asthma questionnaire score had been considered, the prevalence of asthma symptoms in our study (17.3%) would have been similar to that reported in the literature.

We found a positive association between BMI, FVC, and FEV_1_. Malnutrition was negatively associated with both FVC and FEV_1_, whereas obesity was positively associated with FVC only. BMI was used in the multiple regression analysis because there was a significant difference in mean BMI between locations, and BMI was consistently associated with both FVC and FEV_1_ in the univariate analysis. It is well established that malnutrition is associated with worse lung development in childhood [24,25,26], but data on obesity and lung function in children and adolescents are conflicting [27]. BMI is not a good measure of body fat content in children and adolescents [28]. Furthermore, it has been hypothesized that lean body content is positively associated with lung function, whereas body fat content is negatively associated with lung function in children and adolescents [29]. Thus, our findings should be interpreted with caution.

Our study has several strengths. It is one of the few studies that has evaluated the effect of air pollution on lung function in Brazilian children and adolescents. Most Brazilian studies evaluated the short-term effects of air pollution on lung functions in children and adolescents, which were measured only via peak expiratory flow [30,31,32]. A few other studies evaluated other lung function parameters, such as FVC, FEV1, FEV1/FVC, and lung function disorders [33,34,35]. Our study provides further data on the influence of air pollution on lung functions in Brazilian adolescents.

Our study is unique because it evaluated lung functions in a population exposed to silica-rich PM_10_. Some studies evaluated populations living in the vicinity of high-silica-emitting industries and found evidence of several cases of non-occupational pneumoconiosis [8,9,10]. However, these studies did not evaluate the effects of air pollution on lung function in the healthy population, which might be impaired prior to the development of pneumoconiosis. Our study provides evidence that exposure to silica-rich PM_10_ is associated with lower lung function. It is uncertain whether the effects of exposure to silica-predominant particulate matter are similar to those of exposure to traffic-related or biomass-burning-related particulate matter at the same concentration levels [36]. Fuel and biomass burning may produce smaller particles, such as ultrafine particles, that may cause greater damage to human health [37].

Our study had some limitations. First, we were unable to obtain year-long PM_10_ monitoring data for SP. Chosen as a control location (i.e., with low levels of air pollution), SP is a tourist resort town [38,39] without industrial parks or other known sources of air pollution and, thus, was not considered for long-term air pollution monitoring by CETESB. Our brief (48-h) PM_10_ monitoring period in SP may not reflect the actual PM_10_ concentration levels throughout longer time periods, but they are assumed to be lower than the PM_10_ levels of RC and SG. In addition, no air pollutants other than PM_10_ were monitored in RC during 2018 [17], which limited further analysis of air pollution effects in the study subjects.

Second, we did not collect data on variables that might be associated with worse lung development during childhood, such as socioeconomic conditions, birth weight, prematurity, early childhood respiratory diseases, maternal smoking during pregnancy, and secondhand smoking [25,40,41]. We tried to decrease possible socioeconomic biases by inviting only subjects studying at public schools in locations with similar municipality-adjusted human development indices [42].

Third, we did not collect race/ethnicity data. The debate on race/ethnicity in Brazil is complex: government-based profiling relies preferably on skin color (white, brown, black, yellow, and indigenous) instead of ethnicity or origin, and many Brazilians describe themselves as having a skin color other than official profiling options [43]. This classification also does not fully match the GLI options for ethnicities (Caucasian, African-American, Southeast Asian, Northeast Asian, and mixed) [13]. Furthermore, the GLI suggests the use of Caucasian reference data in non-black, non-East Asian, and non-indigenous South Americans [13]. As we did not have race/ethnicity data, we used GLI Caucasian reference data as the standard; thus, we might have misclassified lung function predictions in some participants.

## 5. Conclusions

Exposure to higher PM_10_ concentrations was found to be associated with lower lung function parameters in public high school students in a region of ceramic industrial parks, in which silica particles were predominant in the environmental PM_10_ composition.

## Figures and Tables

**Figure 1 ijerph-20-06964-f001:**
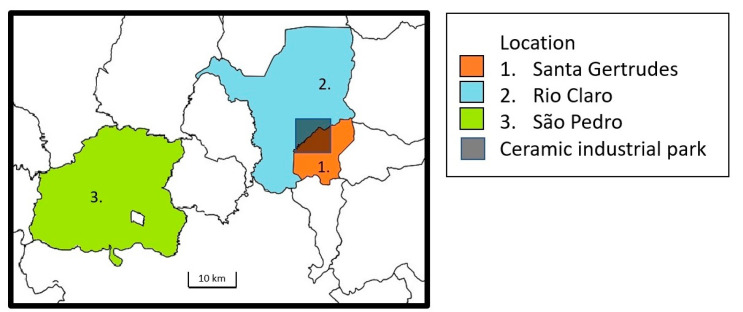
Study location.

**Figure 2 ijerph-20-06964-f002:**
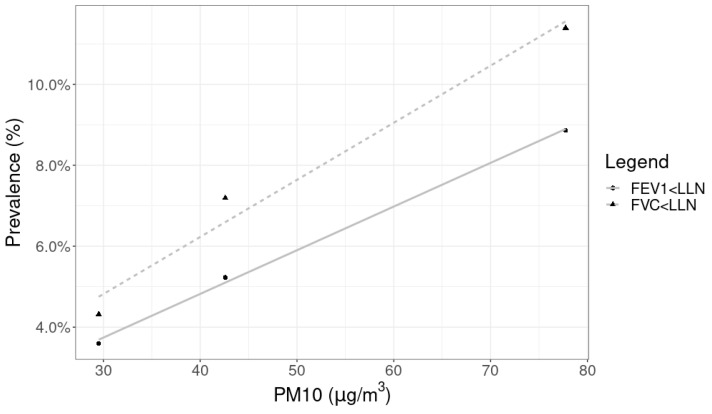
Scatter plot and linear regression of prevalence of the lung function parameter below the lower limit of normality and mean PM_10_. FEV_1_, forced expiratory volume in the first second; FVC, forced vital capacity; LLN, lower limit of normality; PM_10_, particulate matter with an aerodynamic diameter < 10 µm.

**Table 1 ijerph-20-06964-t001:** Characteristics of evaluated students.

	Santa Gertrudes (SG)	Rio Claro (RC)	São Pedro (SP)
Sample (n)	158	153	139
Female sex	88 (55.70%)	80 (52.29%)	63 (45.32%)
Age (years) *	17.05 ± 0.82	16.90 ± 0.64	17.30 ± 0.69 ^†^
Time living in the same location (years)	15.46 ± 2.20	15.63 ± 2.11	15.84 ± 2.00
Height (m)	1.69 ± 0.09	1.68 ± 0.09	1.68 ± 0.09
Weight (kg)	65.35 ± 14.02	66.07 ± 15.78	69.37 ± 14.03
BMI (kg/m^2^) *	22.60 ± 4.02	23.23 ± 4.70	24.47 ± 5.00 ^††^
Malnutrition ^a^	2 (1.27%)	3 (1.96%)	1 (0.72%)
Obesity ^b,^*	9 (5.70%)	16 (10.45%)	26 (18.70%) ^††^
Current/ex-smokers	16 (13.29%)	9 (7.19%)	15 (17.27%)
AS ≥ 6	30 (18.98%)	25 (16.34%)	23 (16.55%)
Asthma diagnosis ^c^	6 (3.80%)	4 (2.61%)	1 (0.72%)

BMI: body mass index; AS: International Study on Asthma and Allergies in Childhood asthma questionnaire score. ^a^ Childhood malnutrition was defined using a BMI-for-age Z-score ≤ −2. ^b^ Childhood obesity was defined using a BMI-for-age Z-score ≥ +2. ^c^ Asthma diagnosis was defined as AS ≥ 6 plus spirometry with obstructive disorder. * *p* < 0.01. ^†^ *p* < 0.05, pairwise comparison between SG and RC. ^††^
*p* < 0.05, pairwise comparison with SG.

**Table 2 ijerph-20-06964-t002:** Lung function evaluation.

	Santa Gertrudes (SG)	Rio Claro (RC)	São Pedro (SP)
Sample (n)	158	153	139
Mean FVC% *	95.0 ± 11.8	98.8 ± 12.9 ^†^	102.4 ± 13.8 ^††^
Mean FEV_1_% *	95.7 ± 10.4	99.7 ± 12.0 ^†^	103.2 ± 12.0 ^††^
Obstructive pattern	16 (10.13%)	13 (8.50%)	10 (7.19%)
Non-specific pattern	3 (1.90%)	2 (1.31%)	1 (0.72%)
FVC < LLN	18 (11.39%)	11 (7.19%)	6 (4.32%)
FVC < 80% **	18 (11.39%)	9 (5.88%)	5 (3.60%)
FVC < 90% **	54 (34.18%)	40 (26.14%)	20 (14.39%) ^†^
FEV_1_ < LLN	14 (8.86%)	8 (5.23%)	5 (3.60%)
FEV_1_ < 80%	11 (6.96%)	8 (5.23%)	4 (2.88%)
FEV_1_ < 90% *	51 (32.28%)	27 (17.65%) ^†^	18 (12.95%) ^†^

FEV_1_, forced expiratory volume in the first second; FVC, forced vital capacity; LLN, lower limit of normality. * *p* < 0.01. ** *p* < 0.05. ^†^ *p* < 0.05, pairwise comparison with SG. ^††^ *p* < 0.05, pairwise comparison with SG and RC.

**Table 3 ijerph-20-06964-t003:** Mean PM_10_ by location.

Location	Mean PM_10_ (µg/m^3^)	Monitoring Period	Type of Monitor
Santa Gertrudes (SG)	77.75 ± 38.08	1 January 2018–31 December 2018	Fixed
Rio Claro (RC)	42.59 ± 23.46	1 January 2018–31 December 2018	Fixed
São Pedro (SP)	29.52 ± 9.87	6 August 2018–8 August 2018	Portable

PM_10_: particulate matter with an aerodynamic diameter <10 µm. *p* < 0.01 for ANOVA and all pairwise comparisons.

**Table 4 ijerph-20-06964-t004:** Univariate linear regression models.

	FVC %	FEV_1_ %
BMI (per 1 kg/m^2^)	**1.05**	**0.43**
	(0.81; 1.29)	(0.20; 0.66)
	*p* < 0.01	*p* < 0.01
Malnutrition ^a^	**−19.2**	**−14.41**
	(−29.64; −8.76)	(−23.91; −4.91)
	*p* < 0.01	*p* < 0.01
Obesity ^b^	**9.23**	**1.93**
	(5.49; 12.97)	(−1.54; 5.40)
	*p* < 0.01	*p* = 0.28
Ever smoker	**−1.18**	**−0.09**
	(−4.95; 2.59)	(−3.50; 3.32)
	*p* = 0.54	*p* = 0.96
PM_10_ (per 10 µg/m^3^)	**−1.43**	**−1.45**
	(−2.01; −0.85)	(−1.97; −0.93)
	*p* < 0.01	*p* < 0.01
AS ≥ 6	**−1.59**	**−3.57**
	(−4.8; 1.62)	(−6.46; −0.68)
	*p* = 0.33	*p* = 0.02
Asthma diagnosis ^c^	**−2.90**	**−11.10**
	(−10.39; 4.59)	(−17.81; −4.39)
	*p* = 0.45	*p* < 0.01

In this table, the beta linear regression coefficient is expressed in bold, and a 95% confidence interval is expressed between the parentheses. FEV_1_%, forced expiratory volume in first-second percent of predicted; FVC %, forced vital capacity percent of predicted; BMI, body mass index; PM_10_, particulate matter with an aerodynamic diameter < 10 µm; AS, International Study on Asthma and Allergies in Childhood asthma questionnaire score. ^a^ Childhood malnutrition is defined using a BMI-for-age Z-score ≤ −2. ^b^ Childhood obesity is defined using a BMI-for-age Z-score ≥ +2. ^c^ Asthma diagnosis was defined as AS ≥ 6 plus spirometry with obstructive disorder.

**Table 5 ijerph-20-06964-t005:** Multiple linear regression models.

	FVC%	FEV_1_%
BMI (per 1 kg/m^2^)	**0.98**	**0.35**
	(0.74; 1.22)	(0.12; 0.58)
	*p* < 0.01	*p* < 0.01
PM_10_ (per 10 µg/m^3^)	**−** **1.10**	**−** **1.27**
	(−1.65; −0.55)	(−1.79; −0.75)
	*p* < 0.01	*p* < 0.01
Asthma diagnosis	-	**−** **10.2**
		(−16.68; −3.72)
		*p* < 0.01

In this table, the beta linear regression coefficient is expressed in bold, and a 95% confidence interval is expressed between the parentheses. FEV_1_%, forced expiratory volume in first-second percent of predicted; FVC%, forced vital capacity percent of predicted; BMI, body mass index; PM_10_, particulate matter with an aerodynamic diameter < 10 µm.

## Data Availability

The data presented in this study are available upon request from the corresponding author.

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
