# Peer review of "An Evaluation of the Impact of Air Pollution on the Lung Functions of High School Students Living in a Ceramic Industrial Park Zone"

_ijerph, 2023, doi:10.3390/ijerph20216964_

Round 1
Reviewer 1 Report
This manuscript study the impacts of pm exposure near silica manufacturing area on respiratory functions among high school students. The article provides interesting results which are suitable for this journal. There are minor comments to be resolved before publication.
Line 48: map of the study location and industrial park will help to understand results.
Line 60: which year was the study conducted?
Line 62: This section need more rational for the inclusion criteria of 10 years of residence in the area. Did the industrial part is established before 10years, so the participants exposed to the same particulate matter?
Line 96: did the two different PM10 observations were validated? Dusttrack is known to overestimate PM concentration comparing to the reference method (i.e., gravimetric method). Were the PM10 data from the three locations comparable?
Line 109: why 2018 PM10 data was used? when was the evaluation days? how the SG, RC, and SP data are comparable? Silica-related respiratory diseases (e.g., silicosis) have long-lag time (~20 years), thus the authors should consider lag-time analysis.
Table 1: mean and standard deviation for the age and BMI seems not significantly different for the three locations.
Table 5: this table needs more explanation. In the method section, the authors mentioned variables with p<0.01 were selected to build multiple regression models, but malnutrition and obesity were not included without explanation. Excluding obesity is acceptable since BMI was included (still it needs explanation).
Line 254: if this is the case, why the authors separate dataset based on the locations (SG, RC, and SP) to assess the impact of pm source on lung function? Participants from SP would have different exposure-impact ratio to SG/RC.
Author Response
We appreciate the time and effort to review our paper, and we are thankful for the considerations to improve its quality.
- Line 48: As a map was included only in the graphical abstract, we included as well to the body of the article.
- Line 60: The study was conducted in 2018, as it was stated previously, at line 50.
- Line 62: Previous to subject recruitment, we contacted the selected school boards, and in SG, it was brought to our attention that many students (up to 10%) had moved recently (< 3 years before) to SG, coming from rural communities of the Northeast region of Brazil, where levels of air pollution are presumed to be low. As our study aimed to evaluate the long-term effects of air pollution on children’s health, we used this 10-year cutoff to avoid the selection of students exposed for few years in the location.
- Line 96: Unfortunately, we were unable to retrieve annual PM10 data for SP as the location is not monitored by the State Agency (CETESB). We monitored all locations with a portable Dusttrak on evaluation days for 48-72h, but we were unable to compare this data due to different meteorological conditions (i.e., it was raining heavily in RC during the evaluation days, and PM10 was very low, with an average 9ppm during those days). Also, as the average PM10 was monitored only every 6 days in SG and RC by CETESB, we were unable to perform a correlation with the data from Dusttrak.
- Line 109: The 2018 PM10 data was used to match the year in which the study was performed. Evaluation days in each location were added on the results.
- Table 1: we rerun the analysis and the ANOVA test was statistically significant with a p < 0.01, probably due to the low standard deviation in the variables. We are able to provide the raw data should you want to run a separate analysis.
- Table 5: in the multiple linear regression models, we avoided to include variables which would evaluate the same risk factor in different ways. So, we opted to include BMI only instead of obesity or malnutrition since these variables are evaluating the same risk factor; we also avoided to include “asthma diagnosis” and “AS score > 6” concurrently. We included this observation in the methods section for clarification.
- Line 254: to assess the dataset only by location could eliminate the biases associated with PM10 measurement in SP, but it would also reduce the results to Tables 1 and 2 only. Eliminating the “control” location (SP) would produce similar consequences, with a poorer dataset to analyze and limited conclusions.
Reviewer 2 Report
The manuscript aims to investigate the influence of Silica rich PM10 pollutants on school students by processing a cross-sectional study to evaluate lung function of public high school students in SG, RC and São Pedro (SP) (control location), Brazil. The result shows that exposure to higher PM10 concentrations was associated with lower lung function parameters. The conclusion provide an important references to Silica rich PM10 pollution related health studies and have direction function for improving to solve related health problems. There are some suggestions to improve the paper’s qualities as follows.
- The introduction could be more logically written. It is suggested that the situation of the PM and related pollution over the world and their health damage could be introduced first, and then the locality information in this study could be given the second.
- Passive sentences were suggested used instead of subject- predicate sentences.
Author Response
We appreciate the time and effort to review our paper, and we are thankful for the considerations to improve its quality.
- We adjusted the introduction as suggested.
- We adjusted the sentences to passive forms, as suggested.
Reviewer 3 Report
The paper is useful for all researchers in the field of environment and human health, especially, in the field of the impact of air pollution on health. The finding suggests that exposure to higher PM10 concentrations is associated with lower lung function parameters in public high school students in a region of ceramic industrial parks. The authors clearly state the limitation of their study, but for the publication of this study I recommend to incorporated some information. Since they study is based on the fact that the composition of PM10 is predominantly silica, they should give the results about the silica content (%). Also, the QA/QC of the measurements for PM10 (even if the measurements have only been performed for 2 days in August 2018 should be given). The text need carful editing for example in line 22 and 24 the 10 in PM10 needs to be in subscript.
Minor editing of English language required.
Author Response
We appreciate the time and effort to review our paper, and we are thankful for the considerations to improve its quality.
- The silica content of PM10 was added to the text.
- We adjusted the PM10 non-subscript misses
Round 2
Reviewer 1 Report
The authors addressed comments well. I recommend accept as present form.
Reviewer 3 Report
As mentioned in the first review the paper is useful for all researchers in the field of environment and human health, especially, in the field of the impact of air pollution on health. The finding suggests that exposure to higher PM10 concentrations is associated with lower lung function parameters in public high school students in a region of ceramic industrial parks. The authors clearly state the limitation of their study, but the authors have not provide the QA/QC of the measurements for PM10 (even if the measurements have only been performed for 2 days in August 2018 ). Since there is significance of this study in the field of the impact of air pollution on health I recommend to publish the paper.